# Improved Active Disturbance Rejection Control Strategy for LCL-Type Grid-Connected Inverters Based on the Backstepping Method

Zhiru Zhang 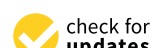 and Wenfang Ding *

Hubei Collaborative Innovation Center for High-Efficiency Utilization of Solar Energy, Power Electronics Team, College of Electrical and Electronic Engineering, Hubei University of Technology, Wuhan 430068, China; 102010310@hbut.edu.cn
* Correspondence: 19861005@hbut.edu.cn

**Abstract:** In the process of grid connection of an LCL inverter, sudden changes in load, high harmonics of the grid voltage, sudden changes in power, and other disturbances often occur. These will cause sharp degradation of the grid-connected power quality, so this paper proposes a new solution to the problem of how to reduce and eliminate disturbances in power quality by using a first-order linear active disturbance rejection control (LADRC) strategy with precise compensation via splitting the total disturbance term. An improved overall direct compensation method for total disturbance was proposed. The use of a subdivision compensation term could avoid the misjudgment arising from the estimation of the overall perturbation by the linear expanded state observer (LESO) within the first few weeks of the feedback when the overall compensation was applied. It aimed to reduce the overshooting caused by the overcompensation of the estimated disturbance term and to shorten the system convergence speed. Backstepping control was introduced to optimize the intermediate quantities of LADRC to estimate the error design outer-loop control law. The controlled quantity tracking the input quantity had excellent characteristics, and could set the desired error range quantity as the purpose of approximation. Therefore, backstepping control was suitable as a feedforward control of the system to preprocess the error in the estimated total disturbance of LESO and feed it into the inner loop improvement LADRC. Secondly, an improved control cascade PWM modulation with a PLL phase-locked loop to regulate the inverter output resulted in the elimination of the effects of internal and external disturbances on the grid-connected current and voltage. Finally, the amplitude–frequency characteristics were analyzed and compared for the trackability and antidisturbance of the improved linear active disturbance rejection controller, showing a good performance of the improved active disturbance rejection. At the same time, comparative simulations were conducted to confirm that the grid-connected current of the LCL inverter could obtain a better stability and grid entry quality in the first-order improved linear active disturbance rejection control.

**Keywords:** LCL grid-connected inverter; linear active disturbance rejection control; estimation error; backstepping control; current quality



## 1. Introduction

In recent years, the proportion of distributed power-generation systems based on solar and wind energy has increased year by year. Energy access is influenced by the environment, often resulting in operation with unknown disturbances such as severe voltage dips and distortions in the grid. The output effect of LCL inverters can be affected by these, and in serious cases, they can go off-grid and affect the stable operation of the grid system [1–4]. However, nowadays, various types of grid-connected devices tend to be diversified, and electromagnetic interference is generated between electronic components [5,6]. Grid conditions are becoming increasingly complex, leading to difficulties in meeting the requirements of

both conventional controls in terms of anti-interference capability and harmonic suppression [7–9]. Many control methods have their own drawbacks, as outlined in [10], or they do not meet the current quality-control requirements of distributed generation grid-connected inverters [11,12] and have not been promoted.

Active disturbance rejection control (ADRC) is applied to grid-connected inverters. It considers disturbances such as current distortion caused by load-side equipment changes, grid-side power glitches, and inverter-side filter uptake as a generalized total disturbance. The control law is designed to compensate for the total disturbance with a good control effect. Due to the more complex structure of a conventional ADRC, the parameters that can be adjusted can number in the dozens, and it is difficult to adjust for better results in engineering [13,14]. To simplify the structure, a linear state error feedback (LSEF) controller was designed. The purpose was to reduce the difficulties in parameter tuning of the nonlinear segmental function control law in a conventional ADRC. An LSEF and a linear extended state observer (LESO) together form the core link of a linear active disturbance rejection control (LADRC) with easy parameter tuning. The parameter tuning of the LADRC [15–17] is normalized to the observer bandwidth and controller bandwidth parameters and thus tuned. However, the control performance is also reduced. A method for modulating fewer LADRC parameters while improving the control performance of the system has become the focus of research. In the literature [18,19], the observer bandwidth was improved by adding an overcorrection link to the LADRC, which improved the observer accuracy. In [20–22], an improved control law for single control of the target by adding decoupling links and designing a time domain and frequency domain combined with LADRC. It had a good suppression effect on the harmonic voltage, but the effect had room for improvement. The authors of [23] improved the dilated state observer to a generalized integral-type dilated state observer by connecting resonant units in parallel to achieve an accurate observation of the frequency. In [24,25], the controlled model was downscaled to maintain the similar transmission characteristics of the higher-order model in the low and middle frequency domains. This was done in order to reduce the complexity of the model in engineering regarding parameter adjustments. The authors of [26–28] designed the LADRC control law to refine the error by splitting the estimation error term, which improved the control law and the quality of grid connection of the LCL-type inverter current.

Based on this, an improved first-order linear active disturbance rejection control strategy based on the grid-connected current stability of LCL inverters is proposed in this paper to counter the grid-connected current quality problem. A conventional LADRC generally compensates for the total disturbance as a whole. It causes a large difference in the initial estimation state, which affects the final convergence time and effect. The objective was to improve the LADRC to take into account the errors that exist between the estimated and actual values of the total perturbation in LESO [29,30]. Firstly, the total disturbance estimate was split by designing the estimation error compensation term to find the exact compensation term required. Secondly, the control law LSEF compensated for it and improved the overall control performance. Lastly, the improved LADRC was combined with backstepping control to achieve the final feedback control.

## 2. Preliminaries and Problem Description

### 2.1. Test System Description

The topology of the LCL inverter is shown in Figure 1. $U_{dc}$ indicates the DC bus voltage of the distributed power supply; $i_{Ma}$, $i_{Mb}$, and $i_{Mc}$ comprise the three-phase grid-side current; $u_{Ma}$, $u_{Mb}$, and $u_{Mc}$ are the load voltages per phase on the network side; $i_{La}$, $i_{Lb}$, and $i_{Lc}$ comprise the three-phase inverter-side current; $u_a$, $u_b$, and $u_c$ are the voltages of the inverter circuit from the center of the three bridge arms to the load; $L_1$ is the inverter-side filter inductor; $L_2$ is the grid-side filter inductor; and $C$ is a Y-connected three-phase filter capacitor.

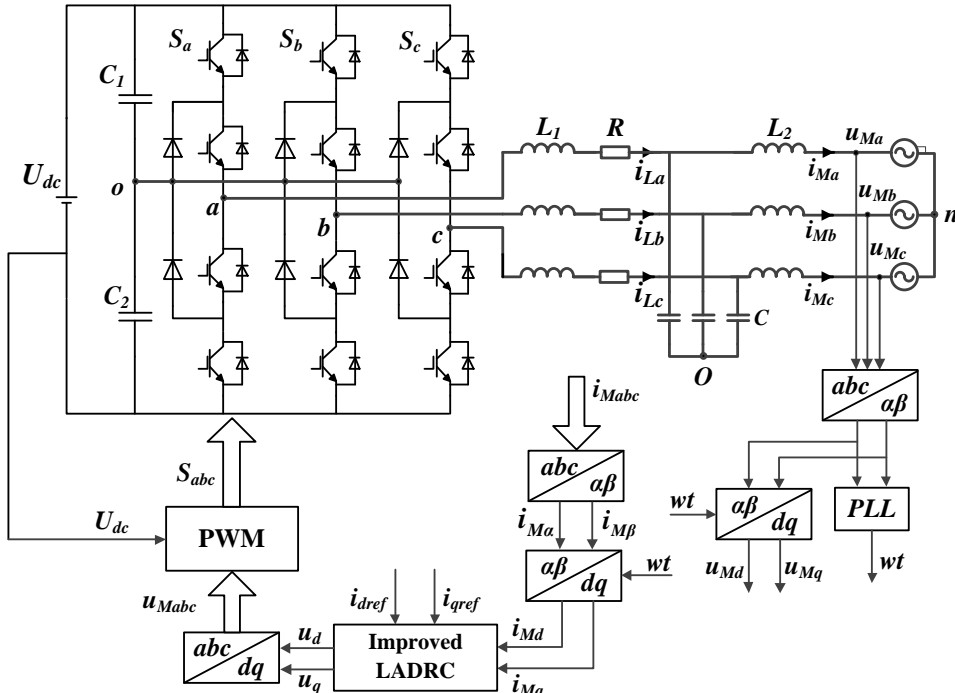

**Figure 1.** LCL grid-connected inverter topology.

The transfer function of the LCL inverter for the current $i_M(s)$ on the three-phase load side and the voltage $u_d(s)$ at the output of the inverter circuit was downscaled according to [31]. The spectral characteristics before and after the degree reduction were further compared and analyzed. The feedback control acted in the low-frequency band and the introduction of the ADRC controller did not require an accurate controlled model; therefore, in order to simplify the complexity of the controller in engineering practice and to not degrade the controller effect, the inverter model was chosen to be processed in a reduced order.

The *dq*-axis model of the reduced-order LCL inverter with capacitor current is:

$$\frac{\mathrm{d}}{\mathrm{d}t}\begin{bmatrix} i_{Md}(t) \\ i_{Mq}(t) \end{bmatrix} = \frac{1}{L_1 + L_2}\begin{bmatrix} u_d(t) \\ u_q(t) \end{bmatrix} + \begin{bmatrix} 0 & \omega_0 \\ -\omega_0 & 0 \end{bmatrix}\begin{bmatrix} i_{Md}(t) \\ i_{Mq}(t) \end{bmatrix} - \frac{1}{L_1 + L_2}\begin{bmatrix} u_{Md}(t) \\ u_{Mq}(t) \end{bmatrix} \tag{1}$$

where $\omega_0$ is obtained by the phase-locked loop for the fundamental frequency of the grid voltage; under the *d*-axis and *q*-axis components of the *dq* coordinate system, $u_d$, $u_q$ is the inverter side output voltage; $i_{Md}$ and $i_{Mq}$ comprise the three-phase load-side current; and $u_{Md}$ and $u_{Mq}$ comprise the three-phase load-side voltage.

### 2.2. Constructing the LCL Inverter State Equation

There were also load-side equipment operating disturbances or unknown disturbances $d_w(t)$ due to faults in the measurement process. Taking the design of the d-axis first-order LADRC controller as an example, the LCL inversion differential equation after reduction to the first order is:

$$\dot{i}_{Md}(t) = \frac{u_d(t)}{L_1 + L_2} + \omega_0 i_{Mq}(t) - \frac{u_{Md}(t)}{L_1 + L_2} + d_w(t) \tag{2}$$

This is because the inverter equation of state (Equation (2)) under complex operating conditions includes three terms in addition to the first term. They are internal disturbances such as changes in filter inductor parameters, electromagnetic interference from inverter-side components, and unknown external disturbances. The internal and external

disturbances in the inverter model are denoted by $f_{ab}(t)$, which is uniformly defined as the total system disturbance as:

$$f_{ab}(t) = \omega_0 i_{Mq}(t) - \frac{u_{Md}(t)}{L_1 + L_2} + d_w(t) \tag{3}$$

Therefore, the differential equation for the inverter load-side current under complex operating conditions can be further expressed as follows:

$$\dot{i}_{Md}(t) = \frac{u_d(t)}{L_1 + L_2} + f_{ab}(t) \tag{4}$$

The state variables were set to $z_{d1} = i_{Md}$ and $z_{d2} = f_{ab}$, where $z_{d2}$ is the expansion state proposed in the linear active disturbance rejection control. Let $h_{ab}$ be the first-order derivative of $f_{ab}$, so that $b_0 u_d(t) = u_d(t)/(L_1 + L_2)$; based on Equation (2), the LCL inverter differential equation is constructed as the state space equation as:

$$\begin{cases} \dot{z}_{d1}(t) = z_{d2}(t) + b_0 u_d(t) \\ \dot{z}_{d2}(t) = \dot{f}_{ab}(t) = h_{ab}(t) \\ z_{d1}(t) = i_{Md}(t) \end{cases} \tag{5}$$

*2.3. Design of Second-Order LESO*

Here, the observations of $z_{d1}$ and $z_{d2}$ corresponding to the expansion observer LESO were defined as $\widehat{z}_{d1}$ and $\widehat{z}_{d2}$, the observations corresponding to the load-side currents $i_{Md}$ and the total disturbance $f_{ab}$ of the inverter grid-connected system, respectively. Since in the state quantity $\beta_2(z_{d1}(t) - \widehat{z}_{d1}(t)) \gg h_{ab}(t)$ of the observed value $\widehat{z}_{d2}$ in LESO, the dilated state observer LESO is obtained by Equation (5) as:

$$\begin{cases} \dot{\widehat{z}}_{d1}(t) = \widehat{z}_{d2}(t) + \beta_1(z_{d1}(t) - \widehat{z}_{d1}(t)) + b_0 u_d \\ \dot{\widehat{z}}_{d2}(t) = \beta_2(z_{d1}(t) - \widehat{z}_{d1}(t)) \end{cases} \tag{6}$$

where $\beta_1$ and $\beta_2$ are parameters in LESO that were optimally tuned to track the LCL inverter state variables in real time.

*2.4. Improved Control Law Design Based on LESO Estimation Error Compensation*

Combining Equations (5) and (6), the error between the state quantity $z_d$ of LESO and the estimated quantity $\widehat{z}_d$ is defined as the estimation error $e_1 = z_{d1} - \widehat{z}_{d1}$, $e_2 = f_{ab} - \widehat{z}_{d2}$ ($f_{ab}$ is the total perturbation), and then Equations (5) and (6) are subtracted to obtain the state space expression using the estimation error $e$:

$$\begin{cases} \dot{e}_1 = e_2 - \beta_1 e_1 \\ \dot{e}_2 = \dot{f}_{ab} - \beta_2 e_1 \end{cases} \tag{7}$$

After finishing by shifting the term in Equation (7) and performing a Laplace transform, we have with respect to the estimation errors $e_1$ and $e_2$:

$$\begin{cases} e_1 = \frac{s}{s^2 + \beta_1 s + \beta_2} f_{ab} \\ e_2 = \frac{s(s + \beta_1)}{s^2 + \beta_1 s + \beta_2} f_{ab} \end{cases} \tag{8}$$

In Equation (8), it can be seen that the only factor influencing the estimation error is the total perturbation $f_{ab}$. Therefore, $e_1$ and $e_2$ are the functions $e_1(f_{ab})$ and $e_2(f_{ab})$ with respect to $f_{ab}$. The approximate compensation is only required for the estimation error of the total perturbation, and its approximate compensation effect determines whether the estimates can accurately follow the state quantities.

The estimated error model of the controlled object based on compensating for the total perturbation is noted as:

$$\dot{z}_{d1}(t) = f_{ab} + b_0 u_d(t) \tag{9}$$

The transfer function of $\widehat{z}_d$ with respect to $z_{d1}$ and $u_d$ in the S domain is obtained by subjecting Equation (6) to a Rasch transform:

$$\begin{cases} \widehat{z}_{d1} = \frac{b_0 s}{s^2+\beta_1 s+\beta_2} u_d + \frac{\beta_1 s+\beta_2}{s^2+\beta_1 s+\beta_2} z_{d1} \\ \widehat{z}_{d2} = \frac{-b_0 \beta_2}{s^2+\beta_1 s+\beta_2} u_d + \frac{\beta_2 s}{s^2+\beta_1 s+\beta_2} z_{d1} \end{cases} \tag{10}$$

Combining the Laplace transform with the Equation (10) ($\widehat{z}_{d2}$), according to the estimation error model of the total disturbance compensated by Equation (9), we get:

$$f_{ab} = \widehat{z}_{d2} + (\beta_1 - \beta_1\beta_2)e_1 + \beta_2 e_2 \tag{11}$$

The design linear state error feedback law is:

$$u_{d0}(t) = k_p(r(t) - \widehat{z}_{d1}(t)) \tag{12}$$

where $k_p$ is the parameter in LSEF, $u_{d0}(t)$ is the output signal of the LSEF link, and $r(t)$ is the grid rated current's given value (the system reference signal).

When substituting $f_{ab}$ from Equation (11) into Equation (13), the equation of state of the controlled object based on the control law at this time is:

$$\begin{aligned} \dot{z}_{d1}(t) &= k_p(r(t) - \widehat{z}_1(t)) + f_{ab} - \widehat{z}_{d2}(t) \\ &= u_{d0} + (\beta_1 - \beta_1\beta_2)e_1 + \beta_2 e_2 \end{aligned} \tag{13}$$

Then, define the estimated error compensation term $f_{ab} - \widehat{z}_{d2}(t)$ in the state equation as $E$:

$$E = (\beta_1 - \beta_1\beta_2)e_1 + \beta_2 e_2 \tag{14}$$

The above equation $E$ is the exact compensation term required in the control law. Since $e_2$ in the estimation error compensation term is the estimation error of the total disturbance, it needs to be converted to $e_1$ uniformly, which is obtained from (8):

$$e_2 = (s + \beta_1)e_1 \tag{15}$$

i.e., there is:

$$E(s) = (\beta_1 + \beta_2 s)e_1 \tag{16}$$

In addition, the estimation error compensation term needs to be eliminated in the final control session, and the linear control law is improved as:

$$u_d = \frac{u_{d0} - \widehat{z}_{d2} - E}{b_0} \tag{17}$$

Figure 2 shows the structure of the modified LADRC, where $r(t) = i^*_{Md}(t)$ is the given value of the d-axis current, $b_0 = 1/(L_1 + L_2)$ is the gain of the linear control law output voltage $u_{d0}$, and $z_{d1}(t) = i_{Md}(t)$ is the final output d-axis component current after eliminating the disturbance on the network-side load. $E$ is able to compensate for the difference between the estimated total disturbance and the actual total disturbance [32].

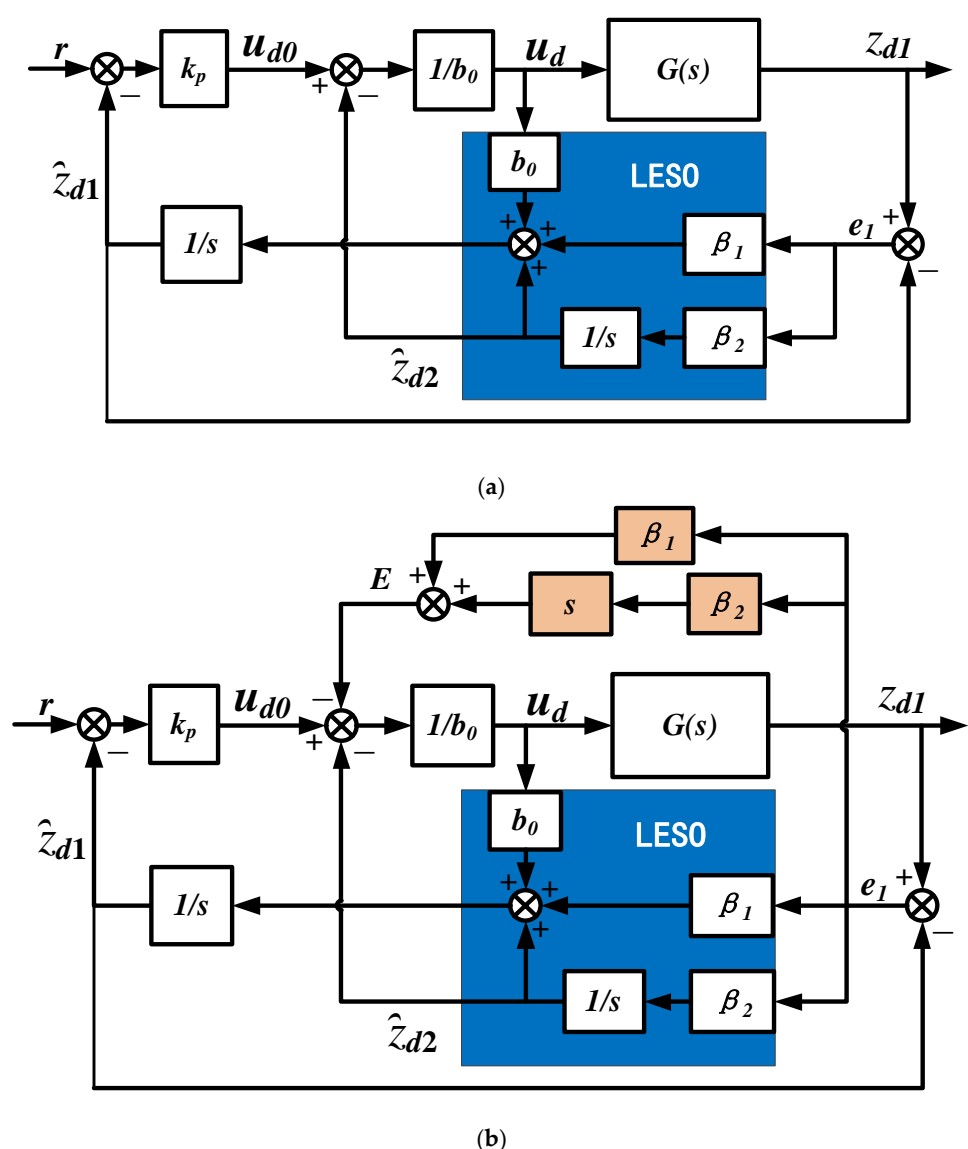

**Figure 2.** Comparison of traditional LADRC structure diagram and improved LADRC structure diagram. (**a**) Traditional LADRC structure diagram; (**b**) improved LADRC structure diagram.

*2.5. Backstepping Outer-Loop Control Law Design*

Backstepping control is often applied in the control strategy of nonlinear systems. The control law is constructed using Lyapunov's second stability theory.

The outer-loop backstepping control constructed the state equation through the inner loop based on the estimation error $e$ of the optimized network-side current $i_{Md}(t)$ versus the estimated current $\widehat{i}_{Md}(t)$. The outer-loop control law was designed to approximate the given target value $\widehat{e}$. The inner loop controlled the output $z_{d1}$ via the target value $\widehat{e}$ after convergence of the outer loop. This inner- and outer-loop design converged the estimation error $e$ to be small enough in the first perimeter compared to the conventional LADRC. There was a good coupling of the internally defined estimation error $e$ as an outer-loop state quantity to design the control law. The target value of the convergence of the design estimation error is $\widehat{e}$. The accuracy requirement was achieved by backstepping to approximate the estimation error $e$. Eventually, the estimated error that converged to the target value was fed into the LADRC control law to create the steps in Section 2.4.

The estimated errors $e_1$ and $e_2$ of the improved inner-loop LADRC control law were used as the spatial state quantities when designing the outer-loop backstepping control

law. According to Equation (7), there are equations of state for $e_1$ and $e_2$, and according to Equation (8), we know that $e_1$ and $e_2$ are transfer functions with respect to $f_{ab}$ as an input. Using $\dot{f}_{ab} = h_{ab}$ as the inputs to $\dot{e}_1$ and $\dot{e}_2$, we constructed the outer-loop state equation from Equation (7) as:

$$\begin{cases} \dot{e}_1 = e_2 - \beta_1 e_1 \\ \dot{e}_2 = h_{ab} - \beta_2 e_1 \end{cases} \tag{18}$$

First, we designed the general control target. The error between the definition $e$ and the target value $\widehat{e}$ of its error is called the target error. The target was designed so that $\delta_1$ converged to 0 in the steady state and so that the equation of state was asymptotically stable with:

$$\delta_1 = e_1 - \widehat{e}_1 \tag{19}$$

According to the definition of the Lyapunov stability, the positive-definite Lyapunov function $V_1(\delta_1)$ was designed as follows:

$$V_1(\delta_1) = \frac{1}{2}\delta_1{}^2 \tag{20}$$

When differentiating (20) and substituting into Equations (18) and (19), we get:

$$\dot{V}_1(\delta_1) = \delta_1(e_2 - \beta_1 e_1 - \dot{\widehat{e}}_1) \tag{21}$$

According to Lyapunov's second stability theory, this must be guaranteed to be negative-definite so that the final state of the system is asymptotically stable. If the first-order-derivative negative-definite condition is satisfied when $\dot{V}_1 = -k_1\delta_1{}^2$ ($k_1 > 0$), set the final target value of $e_2 - \beta_1 e_1 - \dot{\widehat{e}}_1$ to $-k_1\delta_1$ and obtain the target value of $e_2$ ($\widehat{e}_2$):

$$\widehat{e}_2 = \beta_1 e_1 + \dot{\widehat{e}}_1 - k_1\delta_1 \tag{22}$$

Define the target error with respect to $e_2$ and $\widehat{e}_2$:

$$\delta_2 = e_2 - \widehat{e}_2 \tag{23}$$

Similarly, define the new positive-definite Lyapunov function $V_2(\delta_1, \delta_2)$ as follows:

$$V_2(\delta_1, \delta_2) = \frac{1}{2}\delta_1{}^2 + \frac{1}{2}\delta_2{}^2 \tag{24}$$

When substituting (21), (22), and (23) into $\dot{V}_2(\delta_1, \delta_2)$, the derivatives are:

$$\begin{aligned} \dot{V}_2(\delta_1, \delta_2) &= \delta_1\dot{\delta}_1 + \delta_2\dot{\delta}_2 \\ &= \delta_1(\delta_2 + \widehat{e}_2 - \beta_1 e_1 - \dot{\widehat{e}}_1) + \delta_2\dot{\delta}_2 \\ &= -k_1\delta_1{}^2 + \delta_2(\delta_1 + \dot{\delta}_2) \end{aligned} \tag{25}$$

Again, it is necessary to ensure that $\dot{V}_2$ is negative-definite. It is known that the previous term $-k_1\delta_1{}^2$ is negative-definite, and if the final target value of $\delta_2(\delta_1 + \dot{\delta}_2)$ is $-k_2\delta_2{}^2(k_2 > 0)$, then the negative-definite condition of the first-order derivative $\dot{V}_2$ is satisfied when $\delta_1 + \dot{\delta}_2 = -k_2\delta_2$. Substituting Equations (6), (20), (21), and (22), we get:

$$\delta_1 + k_1(e_2 - \beta_1 e_1 - \dot{\widehat{e}}_1) - \ddot{\widehat{e}}_1 - \beta_1(e_2 - \beta_1 e_1) - \beta_2 e_1 + h_{ab} = -k_2\delta_2 \tag{26}$$

The control law of the outer-loop backstepping is obtained by rectifying:

$$h_{ab} = [(k_1 + k_2)\beta_1 - \beta_1{}^2 + \beta_2]e_1 + (\beta_1 - k_1 - k_2)e_2 + (k_1 + k_2)\dot{\hat{e}}_1 + \ddot{\hat{e}}_1 - (1 + k_1 k_2)\delta_1 \tag{27}$$

Figure 3 shows the structure of the backstepping controller designed based on the error $e$ of the optimized grid-side current $i_{Md}(t)$ and the estimated current $\hat{i}_{Md}(t)$. After initially specifying a target value $\hat{e}$ for the estimation error, the backstepping feedback control enabled the estimation error $e$ to approximate the target value. This led to the requirement to optimize the input value of the linear controller (LSEF).

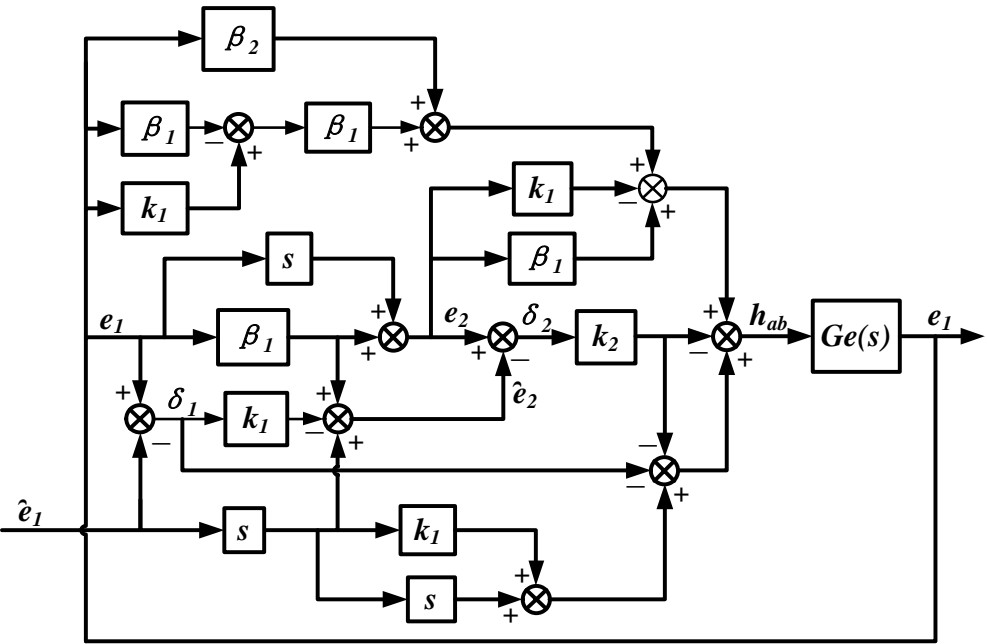

**Figure 3.** Backstepping controller structure diagram.

Thus, the two-step backstepping controller was obtained as:
Figure 4 shows the tracking performance of the backstepping controller after adding random and impulse perturbations to the input waveform. It can be seen that the target values could be tracked faster and better.

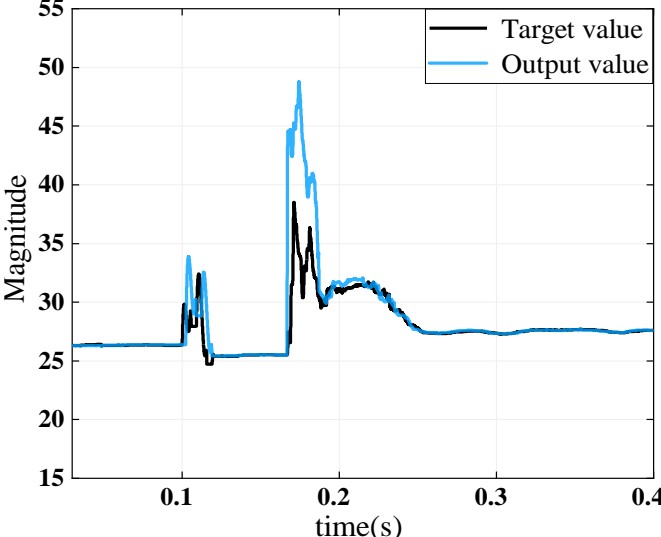

**Figure 4.** Tracking performance of backstepping controller.

Further, coupling the backstepping controller built on the state equation *Ge*(*s*) based on the estimation error *e* and the LCL inverter grid-side current $i_{Md}(t)$ state equation *G*(*s*) improved the control law LADRC control. The BS-LADRC internal-loop and external-loop control structure block diagram is shown in Figure 5.

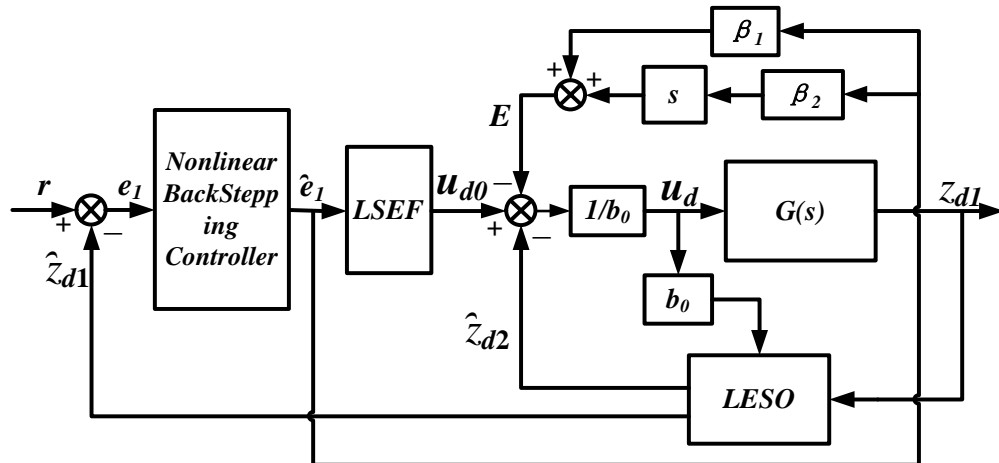

**Figure 5.** BS-LADRC controller structure of internal-loop and external-loop control coupling.

As shown in Figure 5, the input reference signal grid-side current value $i^*_{Md}(t)$ and its waveform were optimized to initially approach the target current value after the output of the outer loop. Thus, the BS-LADRC controller differed from the modified LADRC in Section 2.4 in the input value of the incoming inner loop control law (LSEF). The BS-LADRC controlled the outer-loop output signal to track the set target value, and the input inner-loop control law signal was more stable with fewer errors. The aim was to reduce the convergence time of the inner-loop control and improve the controller's overall robustness.

*2.6. Parameter Design*

Using the bandwidth method to design the observer and controller parameters, the characteristic equation of LESO was obtained as:

$$\lambda(s) = s^2 + \beta_1 s + \beta_2 \tag{28}$$

According to the root trajectory, the poles were all chosen to be at $-\omega_0$. Then, we get $s^2 + 2\omega_0 s + \omega_0^2 = 0$. Similarly, according to the closed-loop control, the poles $-\omega_c$ were chosen to obtain:

$$\beta_1 = 2\omega_0,\ \beta_2 = \omega_0^2,\ k_p = \omega_c \tag{29}$$

At this time, LADRC only needed to reasonably adjust the controller bandwidth $\omega_c$ and the observer bandwidth $\omega_0$ to obtain a good control effect. For most of the common engineering objects, $\omega_0 = (2\sim5)\omega_c$ is generally taken.

**3. Performance Analysis of the BS-LADRC**

The improved LADRC controller transfer function with respect to the input signal *r* and the output signal $z_{d1}$ was obtained using Equations (10), (12), (16) and (17):

$$u_d = \frac{(s + \omega_0)^2 r}{b_0(1 - \omega_0^2)s^2 + \omega_c s} - \frac{[\omega_0^2 s^3 + \omega_0^2 s^2 + (\omega_0^2 + 2\omega_0 \omega_c)s + \omega_0^2 \omega_c]z_{d1}}{b_0(1 - \omega_0^2)s^2 + \omega_c s} \tag{30}$$

The LADRC transfer-function control block diagram was then obtained (see Figure 6).

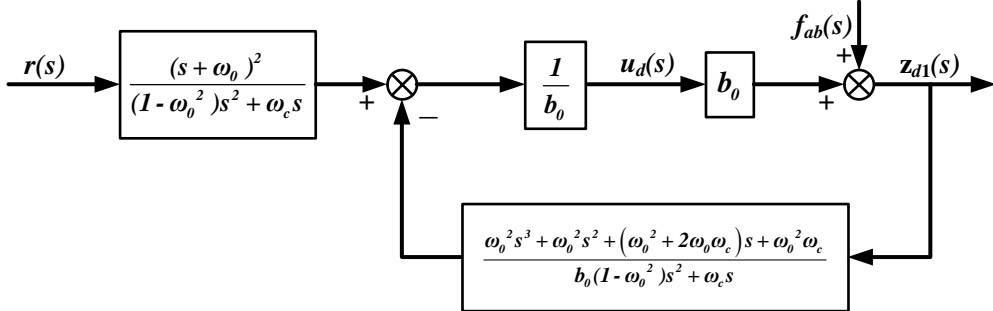

**Figure 6.** Optimal control law LADRC transfer-function-equivalent diagram.

We conducted an overall analysis of the BS-LADRC control strategy, which combined Equations (9) and (30) to obtain the respective transfer functions of the system output $z_{d1}$ with the tracking reference signal $r$ and the total disturbance $f_{ab}$. It represented the tracking performance and antidisturbance performance of the system.

$$\begin{cases} \frac{z_{d1}}{r} = \frac{(s+\omega_0)^2}{s^3+(\omega_c+2\omega_0)s^2+(\omega_0^2+2\omega_0\omega_c)s+\omega_0^2\omega_c} \\ \frac{z_{d1}}{f_{ab}} = \frac{[(1-\omega_0^2)s^2+\omega_c s]}{s^3+(\omega_c+2\omega_0)s^2+(\omega_0^2+2\omega_0\omega_c)s+\omega_0^2\omega_c} \end{cases} \tag{31}$$

Since the LESO had low-pass filtering characteristics, its ability to track disturbances at higher frequencies was limited, and the observed value contained amplitude decay and phase lag compared to the actual value. A comparison of the system tracking performance compared to the transfer function under a conventional LADRC control is shown in Figure 7.

The Bode plot of the tracking performance of the LCL grid-connected system is shown in Figure 8a. The improved first-order self-turbulence controller LCL grid-connected system exhibited stable amplitude–frequency characteristics and a better pass-through performance with low- and medium-frequency signal inputs. Compared with the traditional LADRC phase frequency characteristics, the hysteresis angle was smaller, and the tracking performance showed a very good improvement. With the same bandwidth, the improved LADRC had a larger shear frequency than the conventional LADRC, which had a shorter convergence time and a faster response. As seen in the Bode plot in Figure 8b, the improved LADRC control exhibited more stable amplitude and frequency characteristics and a better pass-through performance with low- and medium-frequency signal inputs.

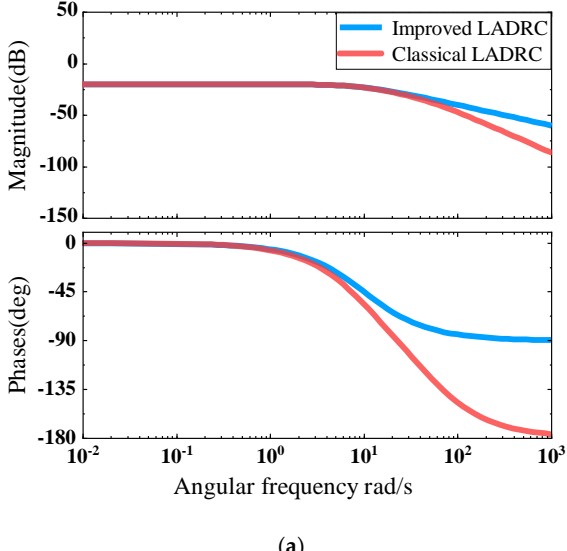

(**a**)

**Figure 7.** *Cont.*

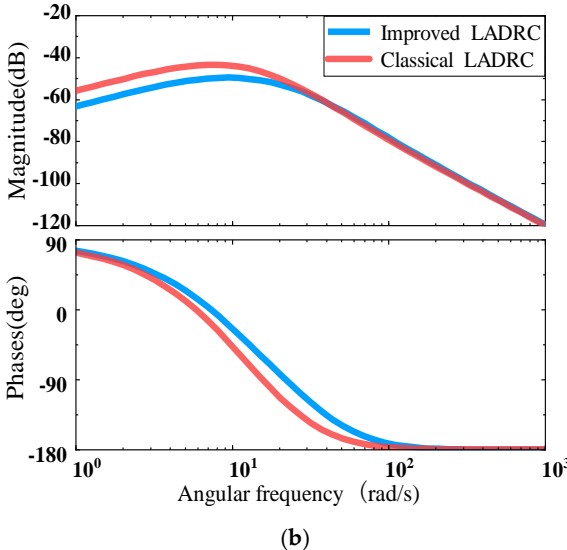

(**b**)

**Figure 7.** Improved LADRC tracking performance and antidisturbance porter diagram. (**a**) Improved LADRC and classical LADRC tracking performance Bode plot amplitude and frequency characteristics comparison analysis; (**b**) improved LADRC and classical LADRC anti-disturbance performance Bode plot amplitude and frequency characteristics comparison analysis.

The adjustment of the parameters affected the antidisturbance control effect of the improved LADRC. It was obtained using Equation (31), and the disturbance term of the system output was related to the two parameters $\omega_0$ and $\omega_c$. The frequency characteristics when $\omega_0 = 20$ and $\omega_c = 20$, 40, or 60 were chosen, as shown in Figure 8a. Taking $\omega_c = 20$ and $\omega_0 = 20$, 40, or 60, the frequency domain characteristic curve could be obtained, as shown in Figure 8b.

In Figure 8, the appropriate increase in the size of $\omega_0$ and $\omega_c$ can be seen. To improve an LADRC to reduce the range of amplitude variation, reducing the size of the phase difference was effective, but its *c* effect was not obvious. Therefore, the main adjustment of the antidisturbance performance was $\omega_0$. The final result was to improve the system's antidisturbance.

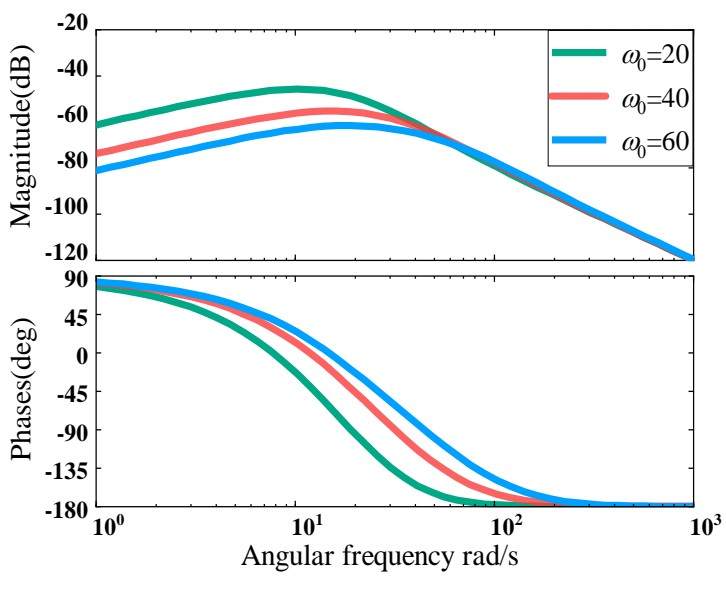

(**a**)

**Figure 8.** *Cont.*

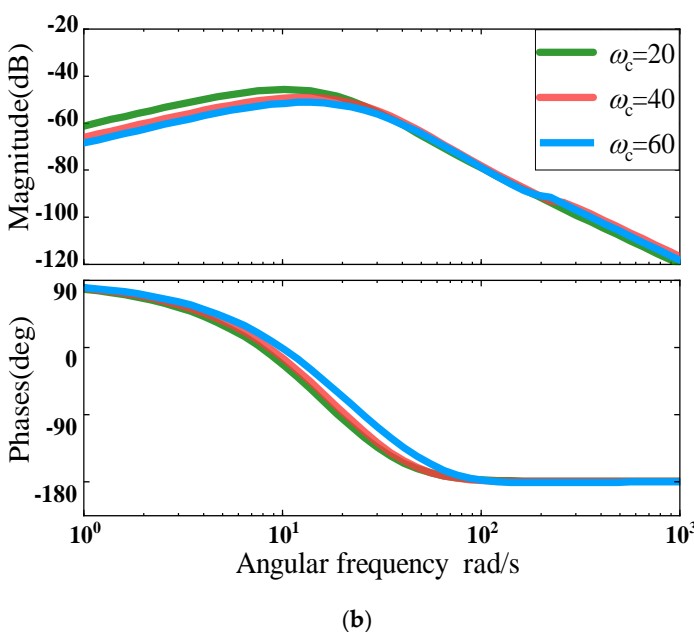

(**b**)

**Figure 8.** Improved LADRC regulation parameters affected the antidisturbance performance porter diagram. (**a**) Comparison of Porter diagrams of anti-interference performance for parameter $\omega_0$ adjustment effects; (**b**) comparison of Porter diagrams of anti-interference performance for parameter $\omega_c$ adjustment effects.

## 4. Simulation Verification

To verify the control effectiveness of the outer-loop backstepping control system in combination with the inner-loop improved LADRC, we established an LCL-type inverter with a combined internal and external loop control and its PWM modulation on a MATLAB/Simulink simulation platform. The simulation modeled a random external disturbance plus a current dip condition on DC side, a sudden change in the disturbance condition on the load side, and high harmonics. Table 1 shows the main parameters used in the modeling.

**Table 1.** Main parameters of system simulation.

| Parameters | Value |
| :---: | :---: |
| DC-side voltage $U_{dc}$/V | 600 |
| Power $P$/kW | 120 |
| RMS grid voltage $u_M$/V | 280 |
| Resonant frequency $f$/Hz | 900 |
| Inverter-side filter inductor $L_1$/mH | 0.6 |
| Switching frequency $f_{sw}$/kHz | 3.2 |
| Grid-side filter inductor $L_2$/mH | 0.3 |
| Filter capacitors $C$/μF | 160 |
| Control gain $b_0$ | 625 |
| Observer bandwidth $\omega_0$ | 1000 |
| Controller initial bandwidth $\omega_c$ | 25 |

For LCL inverters, compound disturbances cause fluctuations in the incoming current, so a better control strategy was needed to better stabilize the output current waveform.

### 4.1. Verifying the Controlled Antidisturbance in Disturbances

The five control strategies; namely, the double-loop PI control, the nonlinear high-gain robust control, the conventional LADRC, the improved LADRC, and the BS-LADRC proposed in the paper, were each used to compare the transient signal landing and interspersion with randomly disturbed waveforms to verify that the BS-LADRC had a better immunity when the system step signal and random disturbance runs were used as input. Figure 9 shows the antidisturbance waveform plots at the moment of random disturbance for the five control strategies. It can be seen that the disturbance fluctuated greatly in the amplitude of the double closed-loop PI control and high-gain robust control. The waveform of the conventional LADRC was relatively better. In this paper, the control effect of the LADRC with an improved control law was significantly improved, but its convergence speed effect was not better. Finally, the BS-LADRC can be seen in the figure as having the best performance. It was able to control the volatility within 1.51% of the input signal and converged quickly, with a 53.59% improvement in the convergence speed compared to the LADRC with an improved control law. Thus, it was verified that the proposed strategy had a faster convergence speed to reduce the tracking time and a better robustness and resistance to external disturbances.

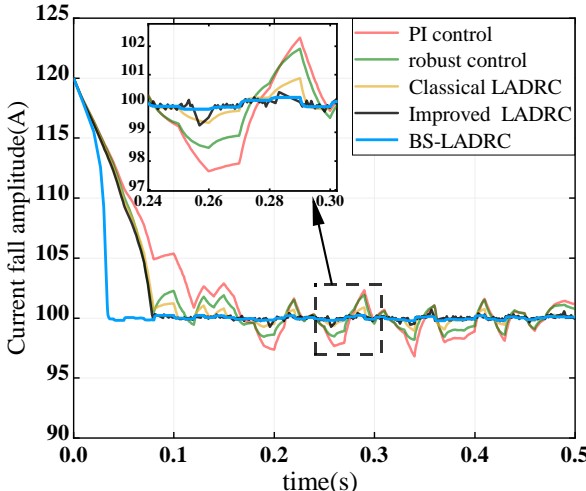

**Figure 9.** Control antidisturbance performance comparison chart.

### 4.2. Verifying the Transient Tracking Performance of Current Disturbances

The total harmonic distortion rate (THD) of the incoming current waveform was analyzed by comparing the output voltage waveform using the double-loop PI control, the conventional LADRC, and the BS-LADRC proposed in this paper. Figure 10 shows the ability of each of the three control strategies to suppress the disturbances. Taking the a-phase current as an example, a random perturbation was added to the initial input signal in the full time domain, and a 20% current pulse perturbation was added at 0.025 s. As can be seen in the figure, the BS-LADRC controller was designed to provide feedforward reverse control of the injected current during the initial transient. It performed preprocessing by estimating the perturbation error so that the tracking phase and amplitude had fewer occurrences of misjudgment, and tracking speed and performance were greatly improved. The double-loop PI control and the conventional LADRC had a large phase difference and an overshoot in the first few cycles of the following curve. When compared to the BS-LADRC, the following interval was short and without overshoot; based on these two points, the BS-LADRC's tracking was better. The proposed BS-LADRC control strategy was significantly better than the previous two in terms of tracking performance, antidisturbance performance, and suppression of harmonic distortion law.

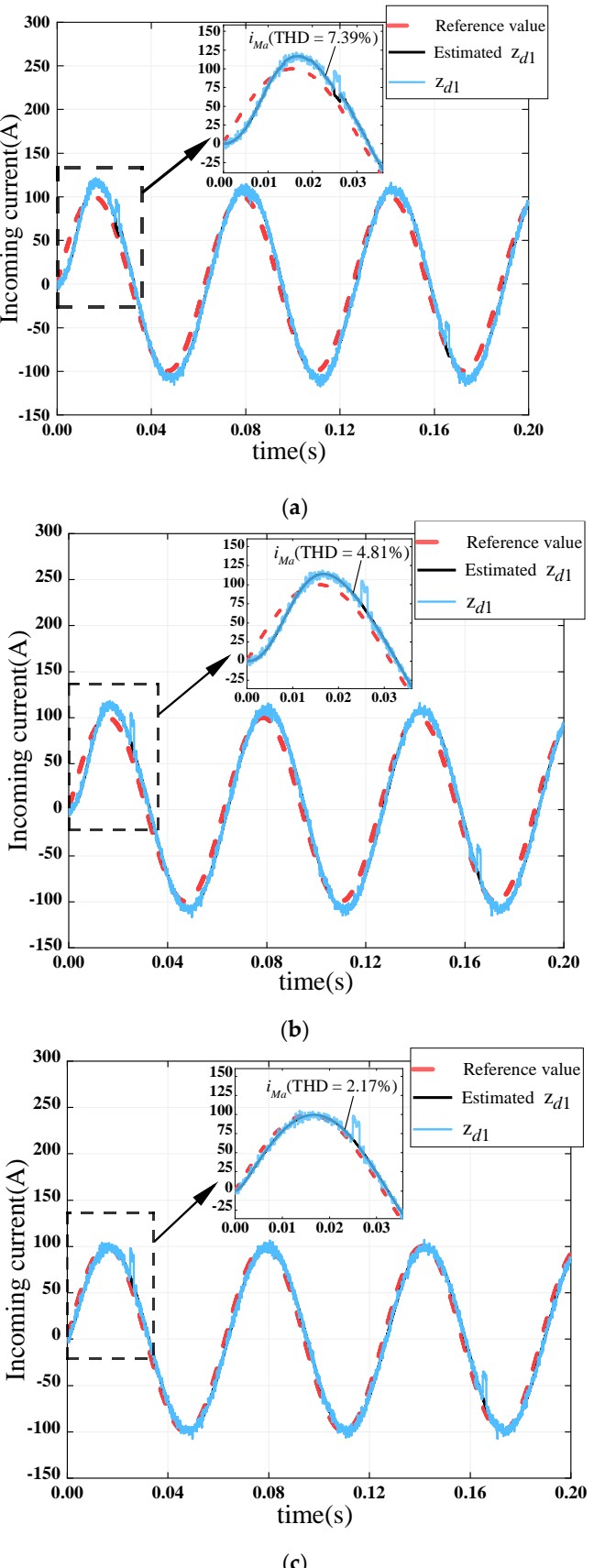

**Figure 10.** Transient tracking performance of current mutation current waveform: (**a**) double-loop PI control strategy; (**b**) classical LADRC strategy; (**c**) BS-LADRC strategy.

### 4.3. Verification of Grid-Side Power-Glitch-Suppression Harmonic Performance

Figure 11 shows the waveform of grid-connected initial current reference $i^*_{Md}(t)$ from 100 A due to a sudden change in power and a sudden increase in current of 195 A and its FFT analysis. Figure 11 shows the steady-state simulation results of the three-phase grid-side currents when the current injected into the grid increased abruptly from 100 A to 195 A due to power surges. Based on the designed improved LADRC control law, the compensation value of the total disturbance was split, and the exact compensation term was found. The small value of the disturbance term of the accurate compensation term E avoided overcompensation in the case of sudden changes in current, and eliminated the disturbance caused by sudden changes in current more accurately and quickly. The simulation tests clearly showed that even under severely abrupt grid conditions, high-quality grid-side currents were maintained with a THD value of 0.83%, which was in line with the specified distortion rate size for grid integration. The current responses reached a steady state after about 0.5 grid voltage cycles. It was clear that the current quickly tracked the new reference value without overshooting. This verified the stability and the ideal transient response of the proposed control scheme. The magnitude of the harmonics was more evident in the FFT results, as shown in Figure 11b, where it can be seen that the current controller did a good job of rejecting the distorted harmonics from the grid. This showed that the BS-ADRC controller limited the distorted harmonics from the grid very well, resulting in a pure output current.

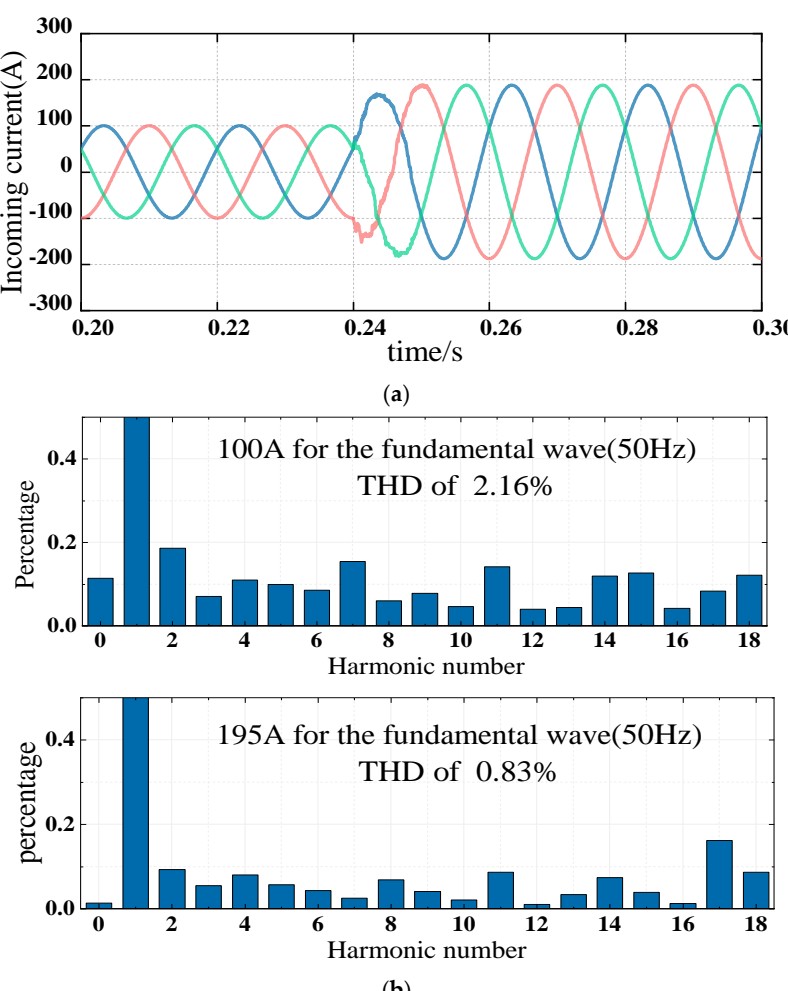

**Figure 11.** Analysis of $i_M$ waveform and FFT of grid current before and after power mutation. (**a**) Sudden power change before and after the incoming current $i_M$; (**b**) FFT analysis of grid-connected currents before and after sudden power changes.

### 4.4. Verification of Grid-Side Voltage Harmonic Distortion Rate's Abrupt Harmonic Suppression Performance

Figure 12 shows the grid voltage when suddenly jittered and mixed with 3.73% higher harmonics, its sudden change before and after the grid current $i^*_{Md}(t)$ to suppress the harmonics, and its FFT analysis. The figure shows the simulation results of the transient grid current response with a distorted grid voltage. The BS-LADRC controlled the transient grid current response in an operating condition at 50 Hz. The moment of the initial phase at which the voltage harmonic distortion occurred showed the most severe transient response. The grid current control performance under the BS-LADRC controller is indicated in the figure. The voltage control began at 0.24 s to inject higher harmonics. The start-up algorithm obtained advance information on the best estimate of the actual grid voltage's magnitude and phase angle. The error between the estimated and actual values was fed into the feedforward control's backstepping control for initial preprocessing to approximate within the desired error range, and the optimized error was fed into the improved LADRC to reduce the possible misjudgment of the estimated value in the first place. The initial transient overcurrent was 106 A; however, it was quickly eliminated by the inverter damping link. In Figure 12b, it can be seen that the output $i_M$ current distortion rate of the grid-connected inverter was controlled to 1.94% compared to the grid voltage of 3.73% after the occurrence of the grid voltage jitter, which still had good power quality.

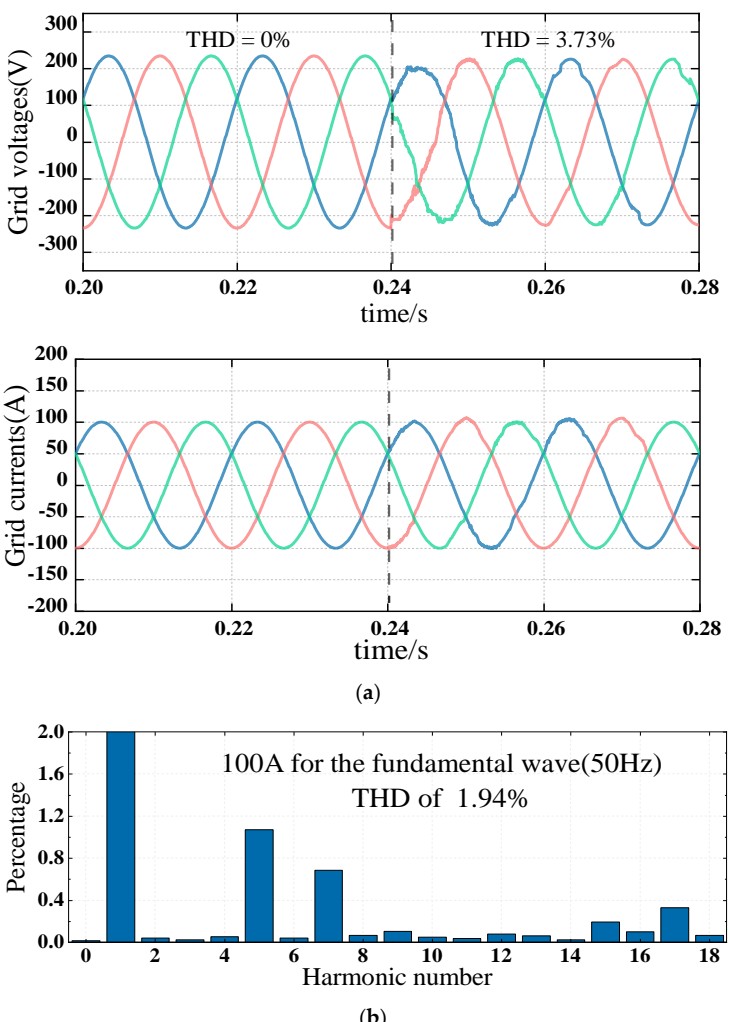

**Figure 12.** Analysis of $i_M$ waveform and FFT before and after grid voltage harmonic wave surge. (**a**) Waveforms of incoming voltage $u_M$ and current $i_M$ before and after the sudden increase in voltage distortion in the power grid; (**b**) FFT analysis of grid-connected current $i_M$ after grid voltage dithering.

### 4.5. Harmonic Suppression Performance of Nonlinear Load Surges in the Power Grid

Figure 13 shows a comparison of the harmonic suppression ability of the improved BS-LADRC (Figure 13a) and the conventional LADRC strategy (Figure 13b) under different load conditions, with the a-phase load current as an example. Among them, 0.2 s to 0.24 s were pure resistive loads, 0.24 s to 0.28 s were switched to rectified nonlinear loads, and after 0.28 s were mixed resistive and rectified loads. A simulation of these load current harmonics and their THD analysis were performed. The total harmonic distortion rate of the output voltage waveform at various loads is indicated in the figure. It can be intuitively seen that the proposed BS-LADRC strategy was more capable of suppressing harmonics under mixed loads with nonlinear loads and resistive and rectified sums compared to the conventional LADRC.

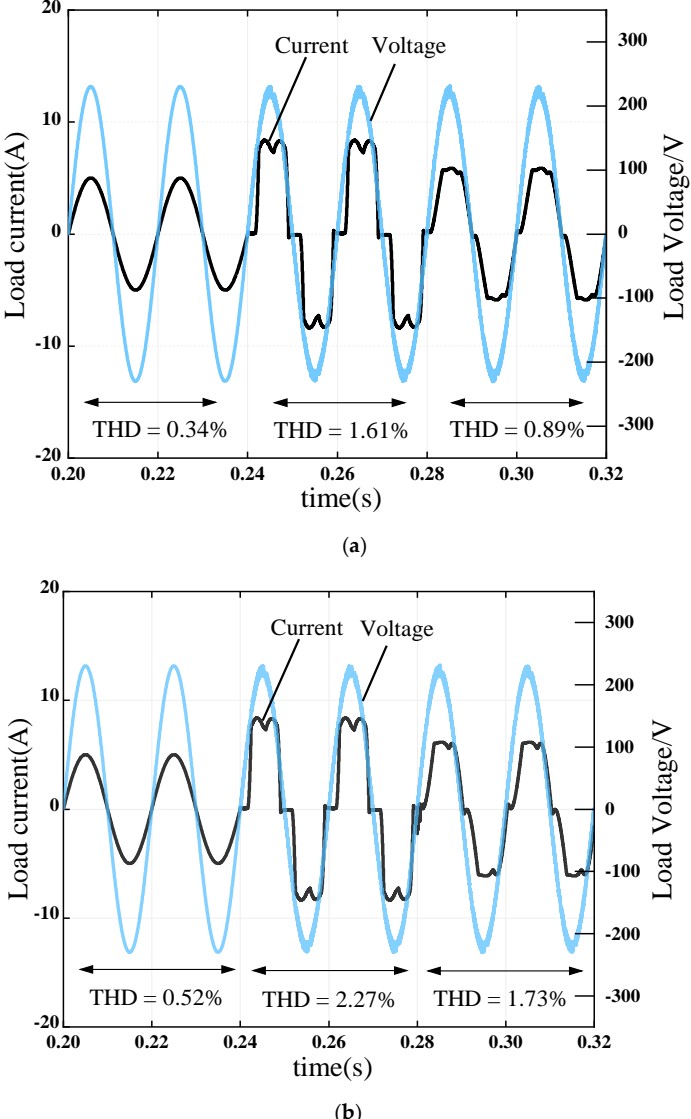

**Figure 13.** Waveforms of voltage and current under different loads. (**a**) Waveforms of voltage and current at different loads in the improved LADRC; (**b**) waveforms of voltage and current at different loads in the classical LADRC.

## 5. Conclusions

In this paper, based on the conventional control of an LCL-type inverter, we combined the idea of self-immunity to generalize the disturbance and observe its disturbance size for compensation. The control law was improved to precisely find the compensation term and to optimize the intermediate state variable *e*. Finally, we proposed the BS-LADRC

strategy. This improved control strategy combined the advantages of a fast convergence of the backstepping control and robustness of the LADRC. It compared with the classical LADRC used for grid-connected current control. The improved controller optimized the case in which the current did not converge fast enough and stored excess current when sudden changes occurred in the system. Its adaptability to the damping link improved the tracking speed and convergence of the feedback, and refined the internal and external perturbations of the system. The system often had resonance, a long-standing type of interference that is difficult to eliminate. There is a need to improve the grid-connected environment when the controller generates high odd harmonics under sudden voltage changes, sudden system power changes, etc. The improved LADRC had better harmonic suppression compared to the classical LADRC. In addition, the control method proposed in this paper was compared with the current cutting-edge LCL inverter current-control strategy [32]. In the exact solution of the total perturbation, the literature provided designs and approximations only for one of them ($e_1$), while the method proposed in this paper solved for the perturbation compensation term *E* more comprehensively while considering the effect of the two quantities of intermediate estimation error ($e_1$ and $e_2$) acting together on *E*. In the case of abrupt changes in the harmonic distortion rate, its THD value in terms of immunity to disturbance showed an improvement of nearly 0.7% compared to that found in the literature. Our study showed that the proposed improved LCL inverter grid-connected current-control strategy could better solve the harsh grid environment problem faced by grid-connected strategies in similar studies we found. It had strong robustness to system dynamic disturbances and harmonics, with a good dynamic and steady-state performance. Subsequent work will focus on the design of the LADRC while taking into account the control delay and modulation delay factors used to improve the immunity and accuracy of LCL inverter systems with active filter hardware.

**Author Contributions:** Z.Z. conceived the main concept of the control structure and developed the entire system; Z.Z. carried out the research and analyzed the numerical data using guidance from W.D.; Z.Z. and W.D. collaborated to prepare the manuscript. All authors have read and agreed to the published version of the manuscript.

**Funding:** This research received no external funding.

**Conflicts of Interest:** The authors declare no conflict of interest.

## Abbreviations

| | |
|---|---|
| $d_w$ | Unknown perturbation |
| $e_1$ | $i_{Md}$ and the error of its estimate |
| $e_2$ | $f_{ab}$ and the error of its estimate |
| $E$ | Real need for compensation |
| $f_{ab}$ | Total disturbance inside and outside the system |
| $i_{La}, i_{Lb}, i_{Lc}$ | Three-phase inverter side *a*, *b*, *c* phase current |
| $i_{Ma}, i_{Mb}, i_{Mc}$ | Grid-connected side *a*, *b*, *c* phase load current |
| $i_{Md}, i_{Mq}$ | Grid-side current under the *d*-axis and *q*-axis components |
| $u_a, u_b, u_c$ | Phase voltage of the inverter circuit from the center of the three bridge arms to the load |
| $u_d, u_q$ | Inverter-side voltage under the d-axis and q-axis components |
| $u_{Ma}, u_{Mb}, u_{Mc}$ | Grid-connected side *a*, *b*, *c* phase load voltage |
| $u_{Md}, u_{Mq}$ | Grid-side voltage under the d-axis and q-axis components |
| $U_{dc}$ | DC busbar voltage |
| $\widehat{z}_{d1}$ | Estimated value of $i_{Md}$ |
| $\widehat{z}_{d2}$ | Estimated value of $f_{ab}$ |
| $\delta_1$ | Error of $e_1$ with its estimated value |

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
