# Peer review of "Improved Active Disturbance Rejection Control Strategy for LCL-Type Grid-Connected Inverters Based on the Backstepping Method"

_electronics, doi:10.3390/electronics11142237_

Round 1

Reviewer 1 Report

Dear Authors

The paper deals with an interesting topic, and the results could be of interest to the readers. However, some relevant points must be attended to improve the manuscript.

1. We must improve the abstract as it does not comply with the relevant information related to contributions.

2. The first sentence in the abstract could be improved. Motivation and the problem must be expressed clearly. The results and contribution must be included.

3. The introduction is wordy. The authors must focus on defining the gap clearly. 

4. The authors must define the novelty and the contributions clearly. Please use bullets to define the different contributions.

5. At the end of the paper, the authors must include a table with abbreviations and symbols.

6. Lines 73-79, not all the terms were included and the font size of variables is incorrect.

7. In subtitle 2.2 the size of the words LCL must be the same.

8. Please try to define each variable in the text or add a table of symbols at the end of the paper.

9. Do not use a combination of uppercase and lowercase in some captions. For example, Figure 3. Backstepping controller Structure diagram.

10. Define correctly Figure 6, do not use only "As shown in Figure 6:"

11. On the first page, line 20, there is a missing name in the citation of the paper. 

12. The size of letters in Figure 10 is not appropriate to read when printing the document.

13. Please, center all the equations according to the paragraph in order to improve the paper's presentation.

14. Correct indentation where the text must give continuity (i.e. where)

15. There are some typing errors and punctuation errors in the text (i.e. strategy, Combining).

16. Improve the analysis reported in Figures 10-13. Those reported miss the importance according to the contributions in the paper.

17. Reduce the conclusion section. Focus on defining first the summary and then the relevant conclusions. I noted the conclusions reported are more related to the methods used and lack of relation according to the required contributions.

18. Include future work at the end of the conclusion section.

19. Delete in the keywords section, the word Terms.

20. Include all the sections that are required in the journal's template (conflict of interest, acknowledgments, etc.).

21. Correct captions in Figure 11. Some are small and with additional spaces. Maybe it will be better to define the text in the main caption and only let (a) and (b). The units should be defined as "Time [s]", "Incoming current [A]". 

22. Identify the lack according to the literature reviewed and use it for defining novelty. 

Author Response

Point 1: We must improve the abstract as it does not comply with the relevant information related to contributions.

Response 1:

For the first point, a detailed description of the contribution points related to the improvement of LADRC performance, which more clearly illustrates the author's research contribution.

Point 2: The first sentence in the abstract could be improved. Motivation and the problem must be expressed clearly. The results and contribution must be included.

Response 2: The first sentence of the abstract, more clearly reflects the purpose of the research and the current problems in the LCL inverter grid connection, and what you have done to contribute.

Point 3: The introduction is wordy. The authors must focus on defining the gap clearly.

Response 3: Revise and reintroduce your innovation, more words to express your difference from other studies.

Point 4: The authors must define the novelty and the contributions clearly. Please use bullets to define the different contributions.

Response 4: The novelty and contribution of the authors' research is highlighted in the abstract section, and the different contributions made by the authors are included at the end of the article.

Point 5: At the end of the paper, the authors must include a table with abbreviations and symbols.

Response 5: A table defining the symbols has been included at the end of the article.

Point 6: Lines 73-79, not all the terms were included and the font size of variables is incorrect.

Response 6: All terms have been included, font size has been corrected.

Point 7: In subtitle 2.2 the size of the words LCL must be the same.

Response 7: Subtitle size has been corrected.

Point 8: Please try to define each variable in the text or add a table of symbols at the end of the paper.

Response 8: Each variable is defined in the text and the symbol table is also added at the end of the text.

Point 9: Do not use a combination of uppercase and lowercase in some captions. For example, Figure 3. Backstepping controller Structure diagram.

Response 9: The case combination problem has been corrected.

Point 10: Define correctly Figure 6, do not use only "As shown in Figure 6:"

Response 10: Defined meaning of Figure 6.

Point 11: On the first page, line 20, there is a missing name in the citation of the paper.

Response 11: Added reference name.

Point 12: The size of letters in Figure 10 is not appropriate to read when printing the document.

Response 12: Graphical layout has been modified.

Point 13: Please, center all the equations according to the paragraph in order to improve the paper's presentation.

Response 13: The equation has been centered.

Point 14: Correct indentation where the text must give continuity (i.e. where)

Response 14: It is not very clear how to change this point, so you may need to explain it more precisely.

Point 15: There are some typing errors and punctuation errors in the text (i.e. strategy, Combining).

Response 15: Typing and punctuation errors have been corrected.

Point 16: Improve the analysis reported in Figures 10-13. Those reported miss the importance according to the contributions in the paper.

Response 16: The analysis reported in Figures 10-13 has been corrected so that the text is more descriptive of the authors' contribution to the study and emphasizes the importance of this work.

Point 17: Reduce the conclusion section. Focus on defining first the summary and then the relevant conclusions. I noted the conclusions reported are more related to the methods used and lack of relation according to the required contributions.

Response 17: Rewrite the conclusion section to provide a systematic summary of the abstract content and a targeted statement of contribution and purpose.

Point 18: Include future work at the end of the conclusion section.

Response 18: Add future work to be done at the end.

Point 19: Delete in the keywords section, the word Terms.

Response 19: Delete redundant parts of keywords.

Point 20: Include all the sections that are required in the journal's template (conflict of interest, acknowledgments, etc.).

Response 20: Add all the sections needed in the journal template.

Point 21: Correct captions in Figure 11. Some are small and with additional spaces. Maybe it will be better to define the text in the main caption and only let (a) and (b). The units should be defined as "Time [s]", "Incoming current [A]".

Response 21: Figure on (a), (b) (s), (A) set as required.

Point 22: Identify the lack according to the literature reviewed and use it for defining novelty. 

Response 22: A comparative description based on the literature reviewed, reflecting the value of the paper's contribution and defining novelty.

Reviewer 2 Report

The reviewer invites the authors to reply and take these comments into consideration as follow:

1.     The abstract has two long sentences; one for the problem and the second sentence shows the purpose of the manuscript with a very short conclusion at the end of the abstract. Please modify the abstract to contain (1) The problem statement (one or two sentences) and purpose of the manuscript (2) The main comparative results in numbers, and (3) The major conclusions.

2.     Avoid long sentences in abstract and through the manuscript to make it easier for the reader to understand them. Please reduce the long sentences (Every sentence should be one or two lines at most) through the manuscript accordingly (There are many long sentences through the manuscript).

3.     The introduction part should be re-organized and cover three parts clearly and sequentially (To make it easy for the reader): 1) Motivation and Incitement, 2) Literature Review and Research Gap, and 3) Contributions and Paper Organization.

4.     In line 353, Figure 11 shows the waveform of grid-connected initial current reference i*Md(t) from 100A due to a sudden change in power and a sudden increase in current of 195A and its FFT analysis. In Figure 11 (b), and 12 (b), the current is 105A which one is correct?

5.     For Figures 11 and 12, adjust the labels to appear all bars/results inside the figure clearly.

6.     In the simulation results (section 4), avoid short paragraphs in sections 4.1 to 4.5. For sections 4.1 to 4.5, it is better to make every section has one paragraph not two short paragraphs.

7.     The declaration of the figures through the main text should be before the figure itself. Please modify for Figures 2, 3, 4, 9, and 10.

8.     Line 281, shown in Fig. 8 (a) not Fig. (a). Also, in line 287, as seen in the Bode Plot Fig. 8 (b) not (b).

9.     In line 388, Figure 13 shows the comparison of the harmonic suppression ability of the improved BS-LADRC (b) and the conventional LADRC (a) strategy under different load conditions 389 with the a-phase load current as an example. It is better to modify and discuss Fig. 13 (a) first for the improved BS-LADRC and then Fig. 13 (b) for the conventional LADRC (Please exchange accordingly).

10.  The conclusions focused on what the authors did in the study. The conclusions should contain and introduce qualitative and quantitative main comparisons results and concluding remarks.

11.   Reference list: The reference writing style didn't match with the electronics journal style. Please modify reference writing style according to (Instructions for Authors) https://www.mdpi.com/journal/electronics/instructions#references

Author Response

Point 1: The abstract has two long sentences; one for the problem and the second sentence shows the purpose of the manuscript with a very short conclusion at the end of the abstract. Please modify the abstract to contain (1) The problem statement (one or two sentences) and purpose of the manuscript (2) The main comparative results in numbers, and (3) The major conclusions.

Response 1:

The abstract has been revised according to the framework of (1) the problem statement and purpose of the manuscript (2) the main comparative results, and (3) the main conclusions;.

Point 2: Avoid long sentences in abstract and through the manuscript to make it easier for the reader to understand them. Please reduce the long sentences (Every sentence should be one or two lines at most) through the manuscript accordingly (There are many long sentences through the manuscript).

Response 2: Revised the long sentences in the article to keep them under two and a half lines long.

Point 3: The introduction part should be re-organized and cover three parts clearly and sequentially (To make it easy for the reader): 1) Motivation and Incitement, 2) Literature Review and Research Gap, and 3) Contributions and Paper Organization.

Response 3: The introduction section has been reorganized in points 1, 2, and 3 as required.

Point 4: In line 353, Figure 11 shows the waveform of grid-connected initial current reference i*Md(t) from 100A due to a sudden change in power and a sudden increase in current of 195A and its FFT analysis. In Figure 11 (b), and 12 (b), the current is 105A which one is correct?

Response 4: Among them, 100A is correct, which has been corrected due to previous oversight.

Point 5: For Figures 11 and 12, adjust the labels to appear all bars/results inside the figure clearly.

Response 5: The labels have been adjusted to show all results in the graph, as requested.

Point 6: In the simulation results (section 4), avoid short paragraphs in sections 4.1 to 4.5. For sections 4.1 to 4.5, it is better to make every section has one paragraph not two short paragraphs.

Response 6: In Chapter 4 Simulation section, each subsection has been changed to a paragraph.

Point 7: The declaration of the figures through the main text should be before the figure itself. Please modify for Figures 2, 3, 4, 9, and 10.

Response 7: The graph declaration has been modified to precede the graph itself.

Point 8: Line 281, shown in Fig. 8 (a) not Fig. (a). Also, in line 287, as seen in the Bode Plot Fig. 8 (b) not (b).

Response 8: Line 281 has been added before (a) (b) Fig.

Point 9: In line 388, Figure 13 shows the comparison of the harmonic suppression ability of the improved BS-LADRC (b) and the conventional LADRC (a) strategy under different load conditions 389 with the a-phase load current as an example. It is better to modify and discuss Fig. 13 (a) first for the improved BS-LADRC and then Fig. 13 (b) for the conventional LADRC (Please exchange accordingly).

Response 9: The improved BS-LADRC and the conventional LADRC have been swapped in position accordingly.

Point 10: The conclusions focused on what the authors did in the study. The conclusions should contain and introduce qualitative and quantitative main comparisons results and concluding remarks.

Response 10: Rewrite the conclusion section to provide a comparative summary of similar research content and a targeted statement of contribution and purpose.

Point 11: Reference list: The reference writing style didn't match with the electronics journal style. Please modify reference writing style according to (Instructions for Authors) https://www.mdpi.com/journal/electronics/instructions#references.

Response 11: The way the references are written has been corrected.

Reviewer 3 Report

The work is well written, clear and easy to read. The diagrams and formulae are also clearly and comprehensively organised. The bibliography and explanation of the problem and the solution are also clear.

It is not clear which calculation system is used to do the calculations in Chapter 3. Chapter 4 then introduces Matlab/Simulink without explaining which calculation system/scheme was used.

It would be better to structure this article according to the logical sequence of the following topics: 1). Matlab model used for the calculation, outline and explanation; 2). validation of the model with various examples and tests comparing with literature or experimental tests; 3). Use of the calculation system and presentation of the results.   

At the least, the calculation system should be better explained.

Apart from these changes, which I consider important, the work is good.

Author Response

I appreciate your compliments and here are my responses to your questions:

Point 1: It is not clear which calculation system is used to do the calculations in Chapter 3. Chapter 4 then introduces Matlab/Simulink without explaining which calculation system/scheme was used.

Response 1: The Porter diagrams in Chapter 3 are obtained by building the transfer function diagrams of the improved controller and the classical controller separately by Simulink. A comparative analysis followed. In Chapter 4, the LCL inverter simulation model is built by simulink and the PWM modulation and PLL phase-locked loop coordination are performed. Different controllers are added and each controller is adapted to the LCL inverter parameters, the width of the modulating wave and the step size of the control operation. If you have a request for a more profound introduction to the computing system, please let me know.

Point 2: It would be better to structure this article according to the logical sequence of the following topics: 1). Matlab model used for the calculation, outline and explanation; 2). validation of the model with various examples and tests comparing with literature or experimental tests; 3). Use of the calculation system and presentation of the results.

Response 2: The framework and logic of the entire article has been revised as requested. Writing in accordance with the given 1). Matlab model for calculation, overview and interpretation; 2). Validation of the model by various examples and tests compared with literature or experimental tests; 3). The use of the computational system and presentation of the results. 

Round 2

Reviewer 1 Report

Dear Authors

Thank you for attending the recommendations.